# ROUTE, INTERPRET, REPEAT:
# BLURRING THE LINE BETWEEN POST HOC EXPLAINABILITY AND INTERPRETABLE MODELS

## ABSTRACT

The current approach to ML model design is either to choose a flexible Blackbox model and explain it post hoc or to start with an interpretable model. Blackbox models are flexible but difficult to explain, whereas interpretable models are designed to be explainable. However, developing interpretable models necessitates extensive ML knowledge, and the resulting models tend to be less flexible, offering potentially subpar performance compared to their Blackbox equivalents. This paper aims to blur the distinction between a post hoc explanation of a Black-Box and constructing interpretable models. We propose beginning with a flexible BlackBox model and gradually *carving out* a mixture of interpretable models and a *residual network*. Our design identifies a subset of samples and *routes* them through the interpretable models. The remaining samples are routed through a flexible residual network. We adopt First Order Logic (FOL) as the interpretable model's backbone, which provides basic reasoning on concepts retrieved from the BlackBox model. On the residual network, we repeat the method until the proportion of data explained by the residual network falls below a desired threshold. Our approach offers several advantages. First, the mixture of interpretable and flexible residual networks results in almost no compromise in performance. Second, the route, interpret, and repeat approach yields a highly flexible interpretable model. Our extensive experiment demonstrates the performance of the model on various datasets. We show that by editing the FOL model, we can fix the shortcut learned by the original BlackBox model. Finally, our method provides a framework for a hybrid symbolic-connectionist network that is simple to train and adaptable to many applications.

## 1 INTRODUCTION

Model explainability is essential in high-stakes applications of AI, such as healthcare. While BlackBox models (*e.g.,* Deep Learning) offer flexibility and modular design, post hoc explanation is prone to confirmation bias Wan et al. (2022), lack of fidelity to the original model Adebayo et al. (2018), and insufficient mechanistic explanation of the decision-making process Rudin (2019). Interpretable-by-design models do not suffer from those issues but tend to be less flexible than Blackbox models and demand substantial expertise to design and fine-tune. Using post hoc explanation or adopting an interpretable model is a mutually exclusive decision to be made at the initial phase of AI model design. This paper aims to blur the line on that dichotomous model design.

The literature on post hoc explainable AI is extensive. The methods such as model attribution (*e.g.,* Saliency Map Simonyan et al. (2013); Selvaraju et al. (2017)), counterfactual approach Abid et al. (2021); Singla et al. (2019), and distillation methods Alharbi et al. (2021); Cheng et al. (2020) are examples of post hoc explainability approaches. Those methods either identify important features of input that contribute the most to the network's output Shrikumar et al. (2016), generate perturbation to the input that flips the network's output Samek et al. (2016), Montavon et al. (2018), or estimate simpler functions that locally approximate the network output. The advantage of the post hoc explainability methods is that they do not compromise the flexibility and performance of the BlackBox. However, the post hoc explainability method suffers from several undesirable significant drawbacks, such as a lack of fidelity and mechanistic explanation of the network output Rudin

(2019). Without a mechanistic explanation, recourse to a model's undesirable behavior is unclear. Interpretable models are alternative designs to the BlackBox model that do not suffer from many of those drawbacks.

Interpretable models also have a long history in statistics and machine learning Letham et al. (2015); Breiman et al. (1984). Several families of interpretable models exist, such as the rule-based approach and generalized additive models Hastie & Tibshirani (1987). Many methods focus on tabular or categorical data and less on high-dimensional structured data such as images. Interpretable models for structured data rely mostly on projecting to a lower dimensional *concept* or *symbolic* space that is understandable for humans Koh et al. (2020). Aside from a few exceptions Ciravegna et al. (2021); Barbiero et al. (2022), the current State-Of-The-Art (SOTA) design does not model the interaction between concepts and symbols, hence offering limited reasoning capabilities and less robustness. Furthermore, current designs are not as flexible as the Blackbox model, which may compromise the performance of such models.

We aim to achieve the best of both worlds: the flexibility of the BlackBox and the mechanistic explainability of the interpretable models. The general idea is that a single interpretable model may not be sufficiently powerful to explain all samples, and several interpretable models might be hidden inside the Blackbox model. We construct a hybrid neuro-symbolic model by progressively *carving out* a mixture of interpretable model and a *residual network*. Our design identifies a subset of samples and *routes* them through the interpretable models. The remaining samples are routed through a flexible residual network. We adopt First Order Logic (FOL) as the interpretable model's backbone, which provides basic reasoning on concepts retrieved from the BlackBox model. FOL is a logical function that accepts predicates (concept presence/absent) as input and returns a True/False output being a logical expression of the predicates. The logical expression, which is a set of AND, OR, Negative, and parenthesis, can be written in the so-called Disjunctive Normal Form (DNF). DNF is a FOL logical formula composed of a disjunction (OR) of conjunctions (AND), known as the "sum of products." On the residual network, we repeat the method until the proportion of data explained by the residual network falls below a desired threshold. The experimental results across various computer vision and medical imaging datasets reveal that our method accounts for the diversity of the explanation space and has minimal impact on the Blackbox's performance. Additionally, we apply our method's explanations to detect shortcuts in computer vision and successfully eliminate the bias from the Blackbox's representation.

## 2 RELATED WORK

**Post hoc explanations** Simonyan et al. (2013); Selvaraju et al. (2017); Smilkov et al. (2017) discuss the post hoc based explanation method using saliency maps to explain a convolution neural network. This method aims to highlight the pixels in the input images that contributed to the network's prediction. Adebayo et al. (2018); Kindermans et al. (2019) demonstrates that the saliency maps highlight the correct regions in the image even though the backbone's representation was arbitrarily perturbed. Additionally, in LIME Ribeiro et al. (2016), given a superpixel, a surrogate linear function attempts to learn the prediction of the Blackbox surrounding that superpixel. SHAP Lundberg & Lee (2017) utilizes a prominent game-theoretic strategy called SHAPLY values to estimate the Blackbox's prediction by considering all the permutations of adding and removing a specific feature to determine its importance in the final prediction. Having the explanations in terms of pixel intensities, they do not correspond to the high-level interpretable attributes (*concept*), understood by humans. In this paper, we aim to provide the post hoc explanation of the Blackbox in terms of the interpretable concepts, rather than the pixel intensities.

**Interpretable models** In this class, the researchers try to design inherently interpretable models to eliminate the requirement for post hoc explanations. In the literature, we find interpretable models in Generalized Additive Models (GAM) Hastie & Tibshirani (1987), or on logic formulas, as in Decision Trees Breiman et al. (1984) or Bayesian Rule Lists (BRL) Letham et al. (2015). However, most of these methods work well in categorical datasets rather than continuous data such as images. Additionally, Chen et al. (2019); Nauta et al. (2021) introduce a "case-based reasoning" technique where the authors first dissect an image in Prototypical parts and then classifies by combining evidence from the pre-defined prototypes. This method is highly sensitive to the choice of prototypes

and the distance metric used to compare with the prototype. So instead of using any prototypes, our proposed method uses human-understandable concepts to build a mixture of interpretable models.

**Concept-based interpretable models**   Recently, Concept bottleneck models (CBM) Koh et al. (2020) broaden the initial aim Lampert et al. (2009); Kumar et al. (2009) to leverage the weak annotations of visual attributes as high-level human-comprehensible concepts Kim et al. (2017) in a fine-grained image classification dataset to predict the concepts from input images. Subsequently, the downstream classifier predicts the class labels from the discovered concepts. Chen et al. (2020) aims to align the axes of the latent space with the concepts of interest. Margeloiu et al. (2021); Mahinpei et al. (2021) demonstrated how the information is encoded in the concept representation. However, they do not explain how the final classifier composes those concepts for prediction. LENs Ciravegna et al. (2021) utilizes a constraint-based approach from Ciravegna et al. (2020a), Ciravegna et al. (2020b) to address this gap by providing explanations in terms of first-order logic (FOL) using the concepts. However, a strong pruning strategy and L1 regularization hinder the learning capability of LENs. E-Lens Barbiero et al. (2022) solves this problem by introducing an attention-driven entropy layer that focuses on the relevant concept for downstream classification. Later, Yuksekgonul et al. (2022) introduces the Post hoc-concept-based model (PCBM) to learn the concepts from the backbone of a trained Blackbox and utilize an interpretable classifier to learn the class labels from the concepts. In addition, they propose the Hybrid-Post hoc-concept-based model (PCBM-h) to replicate the performance of the original Blackbox by fitting the unexplained portion of the Blackbox. However, all these methods either employ a single interpretable by-design model or a surrogate model to explain a Blackbox, failing to apprehend different nuances of explanations per sample. Instead of a single interpretable model for all samples, this paper introduces a mixture of interpretable models that utilize the Blackbox's flexibility and offers local explanation.

## 3   METHOD

**Notation**   Assume we have a dataset $\{\mathcal{X}, \mathcal{Y}, \mathcal{C}\}$, where $\mathcal{X}$, $\mathcal{Y}$, and $\mathcal{C}$ are the input images, class labels, and human interpretable attributes, respectively. $f^0 : \mathcal{X} \to \mathcal{Y}$, is a pre-trained deep neural network (our initial Blackbox model), trained only with supervision from the labels $\mathcal{Y}$. We assume that $f^0$ is a composition $h^0 \circ \Phi$, where $\Phi : \mathcal{X} \to \mathbb{R}^l$ is the image embeddings and $h^0 : \mathbb{R}^l \to \mathcal{Y}$ is a transformation from the embeddings, $\Phi$, to the class labels. We denote the learnable function $t : \mathbb{R}^l \to \mathcal{C}$, projecting the image embeddings to the concept space. The concept space is the space spanned by the attributes $\mathcal{C}$. Per this definition, function $t$ outputs a scalar value representing a concept for each input image.

**Method Overview**   Figure 1 shows an overview of our approach. We aim to iteratively carve out an interpretable model from the given Blackbox model. Each iteration results in an interpretable model (the downward grey paths in Figure 1) and a residual (the straightforward black paths in Figure 1). We start this process with $f^0$ – the initial Blackbox given to us. At iteration $k$, we distill the Blackbox of the previous iteration $f^{k-1}$ with a neuro-symbolic interpretable model, $g^k : \mathcal{C} \to \mathcal{Y}$. We generate explanations in terms of First Order Logic (FOL) from $g^k$ as suggested by Barbiero et al. (2022). We coin the term *residual* $r^k$ to refer to the portion of $f^{k-1}$ that $g^k$ cannot explain, specifically $r^k = f^{k-1} - g^k$. We then approximate $r^k$ with $f^k = h^k(\Phi(.))$. $f^k$ will be the Blackbox for the subsequent iterations and be explained by the respective interpretable models. A learnable gating mechanism, denoted by $\pi^k : \mathcal{C} \to \{0, 1\}$ (shown as the selector in Figure 1) is responsible for routing an input image towards either $g^k$ or $r^k$. The thickness of the lines in Figure 1 represents the samples covered by the interpretable models (grey line) and the residuals (black line). With every iteration, the cumulative coverage of the interpretable models increases, but the residual decreases. We name our method *route, interpret* and *repeat*.

### 3.1   NEURO-SYMBOLIC KNOWLEDGE DISTILLATION

Neuro-symbolic AI is an area of study that encompasses deep neural networks with symbolic approaches to computing and AI to complement the strengths and weaknesses of each, resulting in a robust AI capable of reasoning and cognitive modeling Belle (2020). Neuro-symbolic systems are hybrid models that leverage the robustness of connectionist methods and the soundness of symbolic

reasoning to effectively integrate learning and reasoning Garcez et al. (2015); Besold et al. (2017). Neuro-Symbolic knowledge distillation in our method involves three parts. (1) a series of trainable soft attention scores that *routes* each sample through the interpretable models and the residual networks, (2) A sequence of learnable neuro-symbolic interpretable models, each providing FOL explanations to *interpret* the Blackbox, and (3) *repeating* with Residual networks for the samples that cannot be explained with their interpretable counterparts. We explain each component in detail in what follows.

### 3.1.1 THE SELECTOR FUNCTION

The first step of our method is to *route* an input sample through the interpretable model $g^k$ and residuals $r^k$ using a series of soft attention scores $\pi^k$ for $k \in [0, K]$, where $K$ is the number of iterations. We denote this as the selector in Figure 1. Specifically, at each iteration $k$, the specific selector *routes* a sample $\{x_j, y_j, c_j\}$ towards $g^k$ and $r^k$ with probability $\pi^k(c_j)$ and $1 - \pi^k(c_j)$ respectively. We define the empirical coverage of the $k^{th}$ iteration as, $\zeta(\pi^k) = \frac{1}{m} \sum_{j=1}^{m} \pi^k(c_j)$, the empirical mean of the samples selected by the selector for the corresponding interpretable model $g^k$, with $m$ being the total number of samples in the training set. Thus the total selective risk is,

$$\mathcal{R}^k(\pi^k, g^k) = \frac{\frac{1}{m} \sum_{j=1}^{m} \mathcal{L}^k_{(g^k, \pi^k)}(x_j, c_j)}{\zeta(\pi^k)}, \tag{1}$$

where $\mathcal{L}^k_{(g^k, \pi^k)}$ is the optimization loss used to learn $g^k$ and $\pi^k$ together, discussed in section 3.1.2. For a given coverage of $\tau^k \in (0, 1]$, we solve the following optimization problem

$$\theta^*_{s^k}, \theta^*_{g^k} = \min_{\theta_{s^k}, \theta_{g^k}} \mathcal{R}^k\big(\pi^k(.; \theta_{s^k}), g^k(.; \theta_{g^k})\big) \quad \text{s.t.} \quad \zeta\big(\pi^k(.; \theta_{s^k})\big) \geq \tau^k \tag{2}$$

For $k^{th}$ iteration, $\theta^*_{s^k}, \theta^*_{g^k}$ are the optimal parameters for the selector $\pi^k$ and the interpretable model $g^k$ respectively. In this work, $\pi^k$'s are neural networks with sigmoid activation. At inference time, the concept vector $c_j$ of sample $x_j$ is fed to $g^k$ if and only if $\pi^k(c_j) \geq 0.5$ for $k \in [0, K]$.

### 3.1.2 NEURO-SYMBOLIC INTERPRETABLE MODELS

In this stage for the iteration $k$, we design interpretable model $g^k$ to *interpret* the Blackbox $f^{k-1}$ from the previous iteration $k-1$ by optimizing the following loss function,

$$\mathcal{L}^k_{(g^k, \pi^k)}(x_j, c_j) = \underbrace{\ell\big(f^{k-1}(x_j), g^k(c_j)\big) \pi^k(c_j)}_{\substack{\text{trainable component} \\ \text{for current iteration } k}} \underbrace{\prod_{i=1}^{k-1} \big(1 - \pi^i(c_j)\big)}_{\substack{\text{fixed component trained} \\ \text{in the previous iterations}}} \tag{3}$$

The term multiplied by the loss function $\ell$ in equation 3, is the probability of sample $x_j$ reaching the interpretable model $g^k$, which is the probability of going through the residual for iterations $0$ through $k-1$ $\big($i.e. $\prod_{i=1}^{k-1} \big(1 - \pi^i(c_j)\big)\big)$ times the probability of going through the interpretable model at iteration $k$ $\big($i.e. $\pi^k(c_j)\big)$. Refer to Figure 1 for an illustration. Note that $\pi^1, \ldots \pi^{k-1}$ are learned in the previous iterations and are not trainable at iteration $k$. As each interpretable model $g^k$ specializes in explaining a specific subset of samples (denoted by coverage $\tau$), we refer to it as an *expert* moving forward. We adopted the optimization strategy, discussed in Geifman & El-Yaniv (2019) to optimize equation 2. We discuss the loss function in equation 3 in detail in Appendix A.3. After training, we refer to the interpretable experts of all the iterations as a "Mixture of Interpretable Experts" (MoIE) cumulatively.

### 3.1.3 THE RESIDUALS

The last step is to *repeat* with the residual $r^k$, as $r^k(x_j, c_j) = f^{k-1}(x_j) - g^k(c_j)$, to generate $f^k$, effectively making a new Blackbox for the next iteration $k+1$. Specifically, we train $f^k = h^k\big(\Phi(.)\big)$

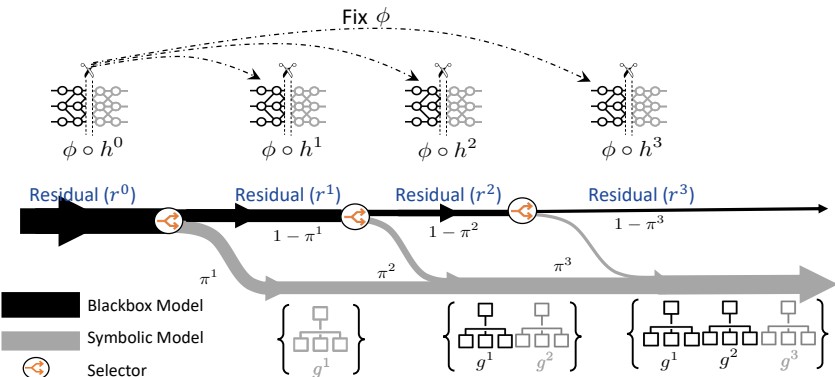

Figure 1: Schematic view of *route, interpret* and *repeat*. At step $k$, the selector *routes* each sample either towards the interpretable model $g^k$ (to *interpret*) with probability $\pi^k$ or the residual (to *repeat* in the further iterations) $r^k = f^{k-1} - g^k$ with probability $1 - \pi^k$. Here, $f^{k-1}$ is the Blackbox of the previous iteration, $k - 1$. If the sample goes through $g^k$, we obtain the explanations for that sample in terms of First Order Logic (FOL). Otherwise, the sample goes through the next step until it either goes through a subsequent interpretable model or reaches the last residual. Note that components in black and grey indicate the fixed and trainable modules in our model.

to match the residual $r^k$. This step is crucial to make $f^k$ specialize over those samples not covered by $g^k$. For the $k^{th}$ iteration, we optimize the following loss function to obtain $f^k$:

$$\mathcal{L}_f^k(\boldsymbol{x_j}, \boldsymbol{c_j}) = \underbrace{\ell\big(r^k(\boldsymbol{x_j}, \boldsymbol{c_j}), f^k(\boldsymbol{x_j})\big)}_{\substack{\text{trainable component} \\ \text{for iteration } k}} \underbrace{\prod_{i=1}^{k} \big(1 - \pi^i(\boldsymbol{c_j})\big)}_{\substack{\text{non-trainable component} \\ \text{for iteration } k}} \tag{4}$$

Notice that the embedding $\Phi(.)$ is fixed for all the iterations. Due to computational reasons, we only update the last few layers of the Blackbox ($h^k$) in order to train $f^k$. At the final iteraion $K$, our method yeilds a MoIE, explaining the interpretable component of the initial Blackbox $f^0$ and the final residual for the uninterpretable component of $f^0$. Algorithm 1 in Appendix A.4 explains the training procedure of our model and extraction of FOL to explain samples locally. Figure 8 in Appendix A.4 shows the architecture of our model at inference.

## 4 EXPERIMENTS

To validate our method, we utilize four datasets - 1) bird species classification using CUB-200 Wah et al. (2011) dataset, 4) animals species classification using Animals with Attributes2 (Awa2) dataset Xian et al. (2018), 3) skin lesion classification using HAM10000 dataset Tschandl et al. (2018), 4) Effusion and Cardiomegaly classification from radiology images using MIMIC-CXR dataset Johnson et al.. First, we demonstrate that MoIE does not hinder the performance of the Blackbox. Second, we conduct qualitative analysis to illustrate the effectiveness of local explanations. Third, we validate that the cumulative coverage of the residuals decreases over successive iterations. Finally, we leverage the Waterbirds dataset Sagawa et al. (2019) to eliminate shortcut bias in vision datasets.

**Training configurations** As a Blackbox, we use ResNet-101 He et al. (2016) and Vision Transformer (VIT) Wang et al. (2021) for the CUB-200 and Awa2 datasets. For HAM10000 and MIMIC-CXR, we utilize Inception Szegedy et al. (2015) and Densenet-121 Huang et al. (2017) as the same. For CUB-200, we employ six experts each for both VIT and ResNet-based Blackboxes. For Awa2, we utilize 4 and 6 experts for ResNet-101 and VIT, respectively. For HAM10000 and MIMIC-CXR (for both Cardiomegaly and Effusion), we use 5 and 3 experts, respectively. Refer to Appendix A.2 and A.7 for details about the dataset and hyperparameters. Furthermore, we only include concepts as input to $g$ if their validation accuracy or auroc exceeds a certain threshold (in all of our experiments, we fix 0.7 or 70% as the threshold of validation auroc or accuracy). For HAM10000, we use the identical "concept bank" as proposed in Yuksekgonul et al. (2022).

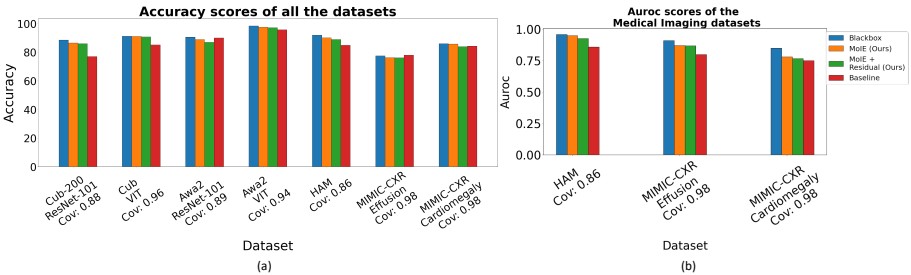

Figure 2: MoIE does not hurt the performance of the Blackbox. From left to right, we compare the (a) accuracy and (b) auroc metrics respectively. For the MoIE (ours), we specify the empirical coverage as *cov* in the figure w.r.t each datasets.

## 4.1 BASELINE

Our baseline is based on the *interpretable by design* principle, consisting of two parts: 1) a concept extractor $\Phi : \mathcal{X} \rightarrow \mathcal{C}$, a mapping from the image to the concept space; and 2) $g : \mathcal{C} \rightarrow \mathcal{Y}$, a mapping from the concepts to the prediction. We train $\Phi$ and $g$ sequentially from scratch with the supervision of the human attributes $\mathcal{C}$ and class labels $\mathcal{Y}$ respectively. The concept extractor $\Phi$ for Resnet, Inception, and Densenet121 includes all layers until the last convolution block. The VIT-based $\Phi$ consists of the transformer encoder block (excluding the classification head). We use the identical configurations for $g$ as described in Appendix A.7 for each dataset.

## 4.2 RESULTS

### 4.2.1 QUANTITATIVE ANALYSIS OF MOIE WITH THE BLACKBOX AND BASELINE

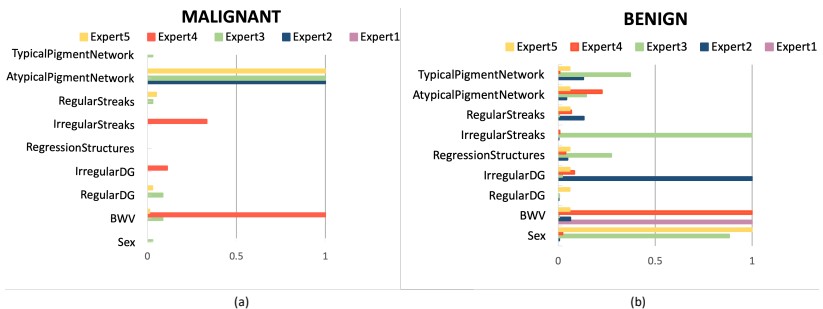

Figure 3: From left to right, for each expert, bar plot showing the importance of a concept for predicting a skin lesion as Malignant and Benign respectively. For example, *atypicalpigmentnetwork* concept plays a crucial role for predicting "Malignant" skin lesion by expert5 as is present in all the local explanations for expert5.

Figure 2 displays the performance of our method using a heldout test set. "MoIE" and "MoIE + Residual" denote the mixture of interpretable experts ($g$) excluding and including the final residual, respectively. Coverage (Cov in the figure) represents the proportion of samples in the test set covered by all the experts throughout iterations. For CUB-200, Awa2, and HAM10000, MoIE achieves similar performance to the Blackbox. For CUB-200 and HAM10000, our method achieves a significant performance gain compared to the baseline. For Awa2, the baseline performs at par with our model. (ResNet-based baseline exceeds ours). Note that all the datasets include noisy concepts except Awa2, making it appropriate for the *interpretable by design* models. In general, VIT-derived experts performed better than their ResNet-based counterparts. The MIMIC-CXR dataset is highly imbalanced, with the positive samples being too few compared to the negative ones. As a result, the baselines for them attain similar accuracy scores as our MoIE model (Figure 2a). Therefore, precision and recall are better metrics for assessing the models' performance on these datasets. To

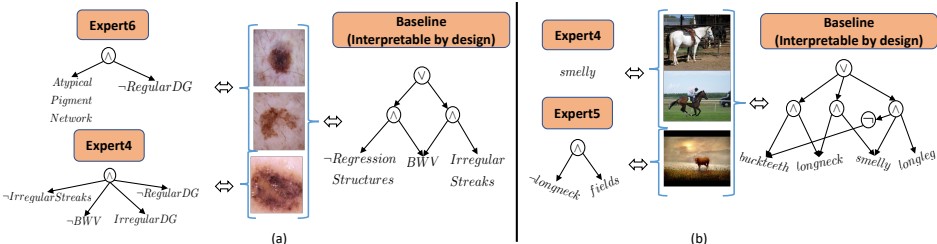

Figure 4: Illustration of flexibility of FOL explanations by MoIE. We report the FOL for different experts and the baseline to identify (a) Malignant lesion and (b) Horse. For example, in Figure (a), the FOL by the baseline composes *BWV*, *RegressionStructures* and *IrregularStreaks* concepts to distinguish all the samples of the skin lesion "Malignant". However in the same Figure (a), expert6 identifies the instances of "Malignant" in the first two rows with *AtypicalPigmentNetwork* and *RegularDG* as the identifying concept. Similarly expert4 classifies the samples of "Malignant" in the third row of Figure (a) using other *BWV*, *IrregularStreaks*, *RegularDG* and *IrregularDG*.

demonstrate this, we plot the auroc in Figure 2b. As it can be seen, MoIE significantly outperforms the baseline in terms of auroc.

### 4.2.2 LOCAL EXPLANATIONS BY MoIE

As shown in E-Lens Barbiero et al. (2022), each expert constructs local FOL per sample, composing the concepts based on their attention weights. Due to space constraints, we report the analysis for the FOL explanation of HAM10000 and CUB-200 datasets here; Appendices A.9.1 and A.8.5 include the results of Awa2 and MIMIC-CXR datasets, respectively. Figures 3a-b exhibit a quantitative analysis of the relative frequency of the concepts used by various experts to correctly classify the "Malignant" and "Benign" skin lesions, respectively. As per Figure 3a, local FOLs' of all samples analyzed by experts 2, 3, and 5 include the concept *AtypicalPigmentNetwork*, proving it to be an essential concept for predicting malignancy. Also, expert4 (the red bars in the Figure 3a) classifies "Malignancy," using multiple concepts, such as *IrregularStreaks*, *IrregularDG*, and *Blue Whitish Veil (BWV)*, whereas expert2 (the blue bar in the Figure 3a) just utilizes *AtypicalPigmentNetwork*.

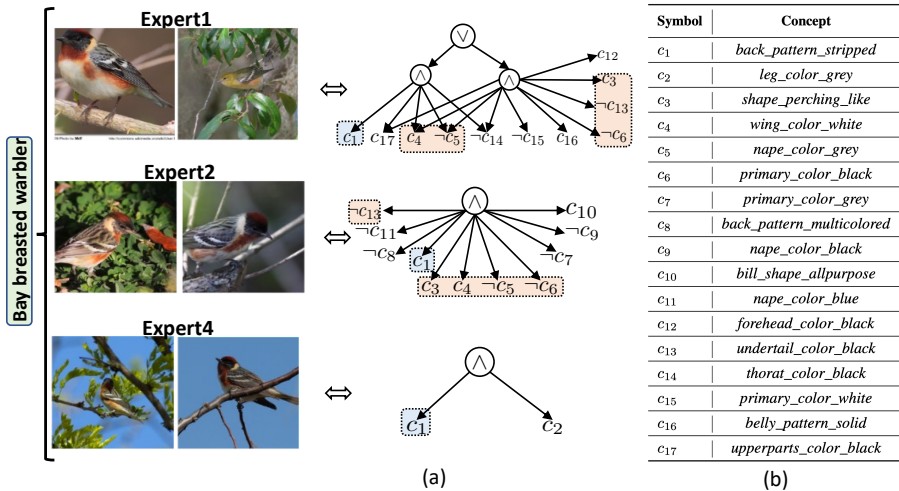

Figure 5: Illustration of class-level or global explanations. We report the global explanation of "Bay breasted warbler" using the VIT-based blackbox by combining (using DNF) the local explanations. (a) Explanations by expert1 (top row), expert2 (middle row) and expert4 (bottom row). Shared concepts across all the experts are highlighted in blue. We highlight the same for experts 1 and 2 in orange. (b) Symbols used in Figure 3a and their corresponding true concept labels.

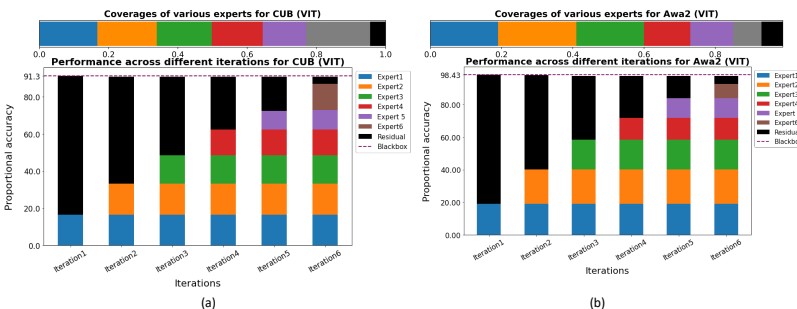

Figure 6: Coverage and performance of each VIT-derived expert and residual for (a) CUB-200 (b) Awa2

Figure 4 compares the FOL explanations by different experts with the baseline. As the baseline includes a single interpretable model $g$, all the FOLs for each sample per class incorporate identical concepts. According to Figure 4a, expert4 relies on *Blue Whitish Veils (BWV)*, *Irregular Streaks*, *Regular* and *Irregular Dots and Globules (DG)* to identify "Malignancy", whereas expert6 utilizes *Rregular Dots and Globules* and *ATypical Pigment Net*. Appendices A.9.1 and A.8.7 display more qualitative FOLs for Awa2 and HAM10000 datasets respectively. MoIE demonstrates similar flexibility in creating the FOL explanation for Awa2. For example, in 4b, the baseline utilizes *longneck*, *smelly*, *longleg* and *buckteeth* to identify all the samples of "Horse". MoIE employs the concepts on a per-sample basis, so different experts adopt different concepts to classify the samples of a particular class. In Figure 4 the expert4 uses *smelly*, but the expert5 uses *fields* and *longneck* to distinguish "Horse".

We perform the same analysis for CUB-200 to formulate a global explanation of a class by combining the different local FOLs per sample using *conjunctive normal form* (CNF) as in E-Lens. Figure 5 depicts such global explanations for the species "Bay-breasted warbler" by three distinct experts. Here we capture the heterogeneity in class-level explanations since each FOL corresponding to an expert has an amalgamation of distinct and shared concepts. For instance *leg_color_grey* is unique to expert4, but *belly_pattern_solid* and *back_pattern_multicolored* are unique to experts 1 and 2, respectively.

### 4.2.3 PERFORMANCE OF EACH EXPERT AND THE RESIDUAL

Figure 6 displays the proportional accuracy of MoIE and the residual for the CUB-200 and Awa2 datasets. Proportional accuracy of each model (experts and/or residuals) is defined as the accuracy of that model times its coverage. Recall that the model's coverage is the empirical mean of the samples selected by the selector. The dashed line in each of the plots indicates the performance of the blackbox, $f^0$, denoting the upper bound of our MoIE models. As shown in Figure 6a, the VIT-derived expert and residual cumulatively achieve an accuracy of approximately 91% for the CUB-200 dataset in iteration 1, where the residual's contribution (black bar) is greater than the expert1's (blue bar). Later iterations reveal that the MoIE's performances cumulatively increase, and the residual worsens. In the final iteration, the MoIE contributes most to performance as they carve out all the interpretable segments from the Blackbox. We observe that the final uninterpretable samples that are routed towards the final residual are "harder" samples for the model to predict, resulting in a lower residual accuracy. Tracing these samples back to $f^0$, we observe that the original blackbox has a similar low performance for predicting these samples. We have shown some of these "hard" images side by side with the ones explained by the experts for CUB-200 in Appendix A.8.8. Also, Table 7 verfies this claim, Per our investigation and as it can be in the images, "hard" examples tend to be blurry or distorted images or ones with birds not fully centered or hard to tell from the background.

As shown in the coverage plot, this experiment reinforces Figure 1, where the flow through the experts gradually becomes thicker compared to the narrower flow of the residual with every iteration. We observe similar trends for the other architectures and datasets. For medical imaging (HAM and MIMIC-CXR) datasets and ResNet-derived experts of vision datasets, refer to figures 12 and 11 in the Appendix A.8.4 and A.8.2, respectively.

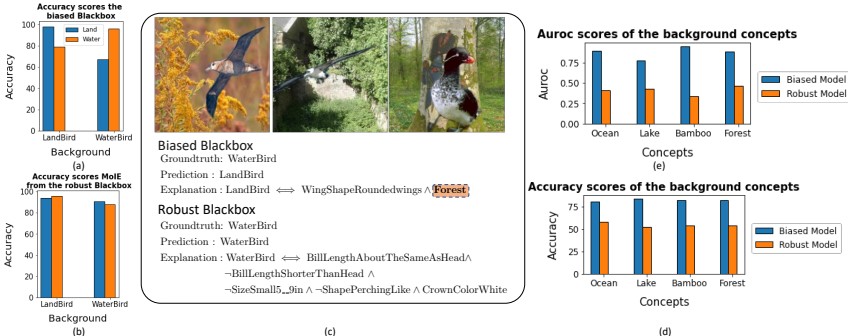

Figure 7: Applying MoIE to eliminate shortcuts. (a) Performance of the biased Blackbox. (b) Performance of final MoIE, trained using the supervision of the robust Blackbox after removing the shortcuts using MDN. (c) Examples of samples and their explanations by the biased (top-row) and robust Blackboxes(bottom-row). (d) auroc and (e) accuracy metrics of the biased Blackbox (when the shortcuts are present) and the robust Blackbox (when the shortcuts are removed using MDN).

### 4.3 APPLICATION IN REMOVAL OF SHORTCUTS

To conduct this experiment, we create the Waterbirds dataset as in Sagawa et al. (2019) by spuriously correlating birds' images from the Caltech-UCSD Birds-200-2011 dataset to the images of land and water background from the Places dataset. Specifically, we use forest and bamboo as the spurious land concepts for landbirds; ocean and lakes as the spurious water concepts for waterbirds. Thus, we have 112 visual concepts (108 concepts from CUB-200 and four new concepts for the background). We aim to identify each bird as a Waterbird or a Landbird. We use ResNet50 (He et al. (2016)) as the Blackbox $f^0$. The Blackbox quickly latches on the spurious background to classify the birds. As a result, the black box's accuracy varies for land-based subsets versus aquatic subsets of the bird species, as shown in Figure 7a. The Waterbird on water is more accurate than on land ((96% vs. 67 % in the orange bar in the Figure 7a). We train $t$ to retrieve the concepts from the biased Blackbox. Next, we train our MoIE to explain the bird species using FOL. As per Figure 7c (middle row), the FOL of a waterbird misclassified as a landbird captures the spurious concept *forest*. Assuming the background concepts as metadata, we aim to reduce the background bias from the representation of the Blackbox using metadata normalization (MDN) Lu et al. (2021). Therefore, we leverage the MDN layers between two successive layers of the convolutional backbone to fine-tune the biased Blackbox. Next, we train $t$, using the embedding $\phi$ of the robust Blackbox. Figures 7d-e compare the auroc and accuracy scores of the spurious background concepts from the embedding $\phi$ of the biased and robust Blackboxes, respectively. The validation auroc (accuracy) of all the spurious concepts retrieved from the robust Blackbox fell well short of the predefined threshold 0.7 (70 %) compared to the biased Blackbox. Finally, we re-train the MoIE distilling from the new robust Blackbox. Figure 7b illustrates that the accuracies of Waterbird on water vs. Waterbird on land become pretty similar (91% -88 %). The FOL from the robust Blackbox does not include any background concepts (Figure 7c, bottom row). Refer to the Figure 9 in Appendix A.6 for the flow diagram of this experiment.

### 5 DISCUSSION & CONCLUSIONS

This paper proposes a novel method to iteratively extract a mixture of interpretable models from a trained Blackbox by utilizing the flexibility of the Blackbox to bridge the gap between post hoc explanation and the construction of interpretable models. The comprehensive experiments on a various datasets demonstrate that 1) our model maintains a similar performance compared to the Black box while making it explainable, 2) the local explanations by our model account for the diversity of explanations of the various samples in a dataset, 3) the coverage of the residuals gradually declines over the iterations. Finally, we employ our method to eliminate the shortcuts in images effectively. We identify the concepts from the representation of the blackbox based on correlation, not causation. So, it is still unclear if the blackbox uses those concepts for prediction. A possible future direction can be the discovering only those concepts causally, used by the blackbox for prediction. In the future we also aim to apply our method to other modalities such as text or video.

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
