# OpenReview forum: "Route, Interpret, Repeat: Blurring the Line Between Posthoc Explainability and Interpretable Models "
_ICLR.cc/2023/Conference — Submitted to ICLR 2023_

### Official Review · Reviewer_gZAS · 2022-10-23

**Confidence:** 3
**Correctness:** 3
**Technical Novelty And Significance:** 2
**Empirical Novelty And Significance:** 2
**Recommendation:** 3

**Clarity, Quality, Novelty And Reproducibility:**

Clarity and quality is acceptable. No code is available (yet) — reproducibility not assessable. I recommend to use anonymised git repositories for future submissions.

**Strength And Weaknesses:**

Strengths: The motivation is clear, the method is described well and the experiments conducted are extensive. The approach has been tested on a variety of datasets and the results have been discussed extensively and from various angles.

Weaknesses: Section 2 on related work seems a bit short, especially the paragraph on concept- based interpretable models could be extended. What methods are used by others and how does the paper at hand differ from those? In line with that, the contributions of the paper in relation to previous work could be highlighted more. Similarly, there are no comparisons done with existing methods (except against the ‘baseline’). If there is a specific reason for that, it was not clear. If not, I would advocate comparing to current state-of-the-art methods for symbolic models for image data. Section 5 (discussion and conclusion) would benefit from a discussion of the strengths and limitations of the presented work. Future research trajectories could be discussed more detailed.

Here are a few additional remarks and questions:

- The readers would benefit from a brief explanation about neuro-symbolic interpretable models in Section 3.1.

- The optimization problem (2) looks highly non-linear and non-convex. The authors have not mentioned difficulty of the problem, neither have they discussed the cases of obtaining multiple or only local solutions. They only propose one method to solve the problem in the appendix. However, this part is crucial as the method relies on the solution of this problem.

- What is a good value for K to stop the algorithm? Is there a rule-of-thumb when to stop?

- What is the effect of validation threshold (fixed to 0.7 in the paper)?

- How would the method extend to non-image datasets?

- What is a concept extractor in Section 4.1?

- There are few typos that I spotted in the current version:
   * In the beginning of Section 3, all $\phi$ functions should be $\Phi$.
   * When referring to a figure, the authors sometimes use "figure #" instead of "Figure #"
   * In equation (3), $c_j$ terms should be boldface

**Summary Of The Paper:**

This paper advocates a hybrid approach between inherent interpretability and post-hoc explainability for predictive models trained on image data. The aim is to blur the line between post hoc explanations of a Black-Box and constructing interpretable models, aiming to keep the high performance and flexibility of blackbox models while providing interpretability. To this end, the authors propose an iterative approach where at each iteration, an interpretable model is distilled from the blackbox model. Samples that cannot be routed through the interpretable model(s) at an iteration are routed through a residual network, which is again a blackbox. The method is repeated on the residual network until until the proportion of data explained by the residual network falls below a desired threshold.

**Summary Of The Review:**

The authors motivate their case and present extensive results. The aim of the proposed approach is described well but there are important missing details. The performance and results of the approach are discussed extensively. The paper could benefit from a more thorough discussion of related work on symbolic models and limitations of the proposed method. There are several typos in the manuscript.

---

> ### Author Response · Authors · 2022-11-12
> **optimization, missing background, clarification**
>
> We thank the reviewer for the constructive feedback. We hope our reply answer the comments. We would be happy to address
> any outstanding issues.
>
> > [...] Section 2 on related work seems a bit short, especially the paragraph on concept-based interpretable models could be extended. What methods are used by others and how does the paper at hand differ from those? In line with that, the contributions of the paper in relation to previous work could be highlighted more. [...]"
>
> We have updated the section accordingly.
>
> > [...] Similarly, there are no comparisons done with existing methods (except against the ‘baseline’). If there is a specific reason for that, it was not clear. If not, I would advocate comparing to current state-of-the-art methods for symbolic models for image data. [...]"
>
> We compare our model with the blackbox (upper bound) and the interpretable by design baseline (lower bound). Our
> interpretable by design approach is based on the SOTA concept bottleneck (Koh et al., ICML 2020). Regarding other
> symbolic models, during the time of the submission there was hardly any symbolic model for image classification.
> Recently Greybox-XAI (Bennetot et al. 2022, available online after we submitted)
> addresses this gap. However, their approach is based on segmentation, whereas we rely on concept attribute annotations.
>
> > [...] Section 5 (discussion and conclusion) would benefit from a discussion of the strengths and limitations of the presented work. Future research trajectories could be discussed more detailed. [...]
>
> Added in red. Please check.
>
> > [...] The readers would benefit from a brief explanation about neuro-symbolic interpretable models in Section 3.1. [...]
>
> We added a short and long description in red in section 3.1 and Appendix A.1 respectively.
>
> > [...] The optimization problem (2) looks highly non-linear and non-convex. The authors have not mentioned difficulty of the problem, neither have they discussed the cases of obtaining multiple or only local solutions. They only propose one method to solve the problem in the appendix. However, this part is crucial as the method relies on the solution of this problem.[...]"
>
> Thank you for the suggestion. We agree with the reviewer that the optimization landscape is non-convex, and providing a
> guarantee for the global optimum is difficult. However, as is the case for general DL, the local optima provide good
> generalization (Asymmetric Valleys, He et al. [Neurips, 2019]; Sharp Minima, Dinh et al. [ICML, 2017]). Please note that
> models being interpretable by design, are trained from scratch. So the optimization procedure for this class of models
> is more complex. However, distillation from the black box is easier for our model to optimize.
>
> > What is a good value for K to stop the algorithm? Is there a rule of thumb when to stop? [...]
>
> We follow two principles to stop the recursive process. 1) Each expert should have enough data to be trained reliably (
> coverage $\zeta^k$). If insufficient samples fall into the expert, we stop the process. 2) If the latest residual (
> $r^k$) is under-performing, it is not a reliable black box to distill. We stop the procedure to avoid degrading the
> overall accuracy. We add a section about stopping criteria in the appendix or in the algorithm block in **red** color.
>
> > [...] What is the effect of validation threshold (fixed to 0.7 in the paper)? [...].
>
> We use this validation threshold to select only those concepts which the intermediate representation $\Phi$ of the
> blackbox predicts accurately. If the blackbox achieves high performance, its representation must learn the different
> semantic concepts accurately as well. We want to leverage only the highly predictive concepts for explaining the
> prediction of the BlackBox.
>
> > [...] How would the method extend to non-image datasets? [...]
>
> Yes, definitely. Given an example of how the symbolic section (FOL) change be modified to anything else like a program
> and the same idea can be applied for NLP or categorical dataset.
>
> > [...] What is a concept extractor in Section 4.1?[...]
>
> The baseline (interpretable by design), have two parts - the concept extractor($\Phi$) and the downstream classifier (g)
> , **trained from sequentially from scratch**. The concept extractor($\Phi$) maps the input images to the high level
> intermediate concepts (c). The downstream classifier (g) aims to classify the class labels from the concepts, extracted
> by the concept extractor ($\Phi$).
>
> > [...] In the beginning of Section 3, all \phi functions should be \Phi. [...]
>
> We have fixed this issue.
>
> > [...] When referring to a figure, the authors sometimes use "figure #" instead of "Figure #" [...]
>
> We have fixed this issue.
>
> > [...] In equation (3), $c_j$ terms should be boldface[...]
>
> We have fixed this issue.
>
> > [...] No code is available (yet) — reproducibility not assessable [...]
>
> Please refer to the GitHub link in the general comment. Also, we listed all the hyperparameters in Appendix A.6.

---

> > ### Author Response · Authors · 2022-12-09
> > **Did we answer your questions?**
> >
> > Dear Reviewer,
> >
> > In the remaining time, we would be thankful if you let us know if the experiment was convincing or if you need different types of experiments/clarification.
> >
> > If satisfied, we would be thankful if you update your score accordingly.

---

### Official Review · Reviewer_1Nhn · 2022-10-24

**Confidence:** 2
**Correctness:** 3
**Technical Novelty And Significance:** 2
**Empirical Novelty And Significance:** 2
**Recommendation:** 5

**Clarity, Quality, Novelty And Reproducibility:**

### novelty
The iterative approach combining several neuro-symbolic interpretable models is novel.

### clarity
The paper is, for the most part well written. Notation is fine. Some additional descriptions of concepts from other work that the authors use would have been nice.

### Reproducability
Pseudo code is given but no github implementations, so reproducibility may be cumbersome.

### Quality
Given that the approach is quite complicated, the motivation seems a bit poor.
Experiments are of good quality. Formulas are a bit hard to understand.

**Strength And Weaknesses:**

### strenghts
- the explanations are relatively easy to read thanks to FOL
- experiments are numerous and well described
- the authors demonstrate that biases can be detected

### weaknesses

- a short explanation of the first order logic is missing
- a short explanation of ``neuro-symbolic interpretable model'' is missing
- no code
- complicated approach, training required
- no comparison with similar concept based explanation methods (like CAV)
- no quantitative analysis of the produced explanations


### questions
- You write in section 3.1.1 that the selector routes a sample with a probability... Is this routing really probabilistic, so that I would potentially get different explanations when explaining the same image twice?
- You are using the original black box classifier in the first iteration, if I understand correctly. Why is the performance in Figure 2 then not equal to the black box?
- why do you not show explanations for the same samples in figure 4?

**Summary Of The Paper:**

The authors propose an approach to explain black box classifiers by iteratively dividing the black-box into an interpretable model and a residual. They use first order logic on concepts for the interpretable models.
A sample input is passed to the system and either routed (by a learnable gating mechanism) to the interpretable model or the residual, until it is interpreted or reaches the last residual.

**Summary Of The Review:**

Overall I'm not quite convinced by the paper. As the procedure is relatively complicated and requires training, the motivation should be a bit clearer. Furthermore, there needs to be more quantitative evaluations and comparisons to state of the art concept based explanations.

---

> ### Author Response · Authors · 2022-11-12
> **evaluation, missing background, clarification**
>
> We thank the reviewer for the feedback and constructive comments. We hope the following reply address the points raised by the reviewer. We would be more than happy to address any other outstanding issue.
>
>
> > [...] a short explanation of the first order logic is missing [...]
>
> We will add a short description of FOL in the paper. In short, FOL is a logical function that accepts predicates (concept presence/absent) as input and returns a True/False output that is a logical expression of the predicates. The logical expression, which is a set of AND, OR, Negative, and parenthesis, can be written in the so-called Disjunctive Normal Form (DNF). Disjunctive Normal Form (DNF) is a FOL logical formula composed of a disjunction (OR) of conjunctions (AND).
>
>
> > [...] a short explanation of ``neuro-symbolic interpretable model'' is missing [...]
>
> We added a short and long description to the manuscript in red in section 3.1 and Appendix A.1 respectively.
>
> > [...] no code [...]
>
> Please refer to the GitHub link in the general comment. Also, we listed all the hyperparameters in Appendix A.6.
>
> > [...] complicated approach, training required [...]
>
> The proposed approach is **not** a posthoc explanation but rather a new interpretable network that is designed on-the-fly (ie during the recursive process). Therefore, training is needed. Each step of the procedure *carves out* an interpretable model (ie FOL expert) to explain a subset of data. Please note that a single expert (eg FOL) is suboptimal for the entire data; this is why the selector ($g^k$) identifies a subset of samples for the expert $k$. This procedure is repeated until the mixture of experts is sufficiently powerful and on par with the initial BlackBox. We would appreciate it if the reviewer provided further feedback regarding their comment about the complexity. We are using SGD and the arch classical vision CNN or ViT. All hyperparameters are specified in the supplementary.
>
> > [...] no comparison with similar concept based explanation methods (like CAV)[...]
>
> CAV is a **posthoc explanation-based** method aiming to quantify the association of a concept to the final prediction of a blackbox. Our aim is completely different. Our goal is to design a new **interpretable network** that uses the concepts and the interaction between them (via FOL) as bottlenecks. We update
>
> >. [...] no quantitative analysis of the produced explanations [...]
>
> Our method is **not** a post hoc explanation method but rather an interpretable approach. There are many quantitive experiments in the paper. The metric we used for evaluation are consistent with the evaluation of the interpretable methods:
>
> - **Does our method compromise accuracy?** Sec 4.2.1 and Figure 2 show that our method outperforms the interpretable baseline (Concept bottleneck model Koh et el. 2020) and does not compromise the performance of the respective blackbox.
>
> - **Is there any sample that does not fit the template of the interpretable method?** There are always samples that need to fit the interpretable template. Our last selector ($g^k$) can identify those. Please refer to section A 7.7 and Table 6 in the supplementary material for details.
>
> - **Is this a helpful method for application?** This is the primary concern about any XAI approach. We showed that our method could fix the shortcut bias. Please refer to section 4.3 for details.
>
> > [...] that the selector routes a sample with a probability... Is this routing really probabilistic [...]
>
> Yes, it is probabilistic in the sense that the **loss function** is the expectation of the latent variable that selects samples. By taking expectation, the random variable is no longer in the loss.
> $$ \ell( x_i ; \theta ) = \mathbb{E}_{\phi} \left[ \mathbb{1}(x_i)  \ell( x_i, g^k ) + (1 - \mathbb{1}(x_i)) \ell(x_i, r^k) \right], $$ where $\mathbb{1}(x)$ is the indicator function selecting sample $x_i$ for the expert $k$ ($g^k$). We did not sample from this expectation to route samples to different experts as it was not the paper's goal.
>
> > [...]  using the original black box classifier in the first iteration, if I understand correctly. Why is the performance in Figure 2 then not equal to the black box [...]
>
> The figure 2 compares the final performance of the carved interpretable network (ie Mixture of experts) with the BlackBox. The goal of the figure is to show that the new interpretable network does not compromise the accuracy of the BlackBox. We are distilling from the BlackBox to a new network, so they are not the same.
>
> > [...] why do you not show explanations for the same samples in figure 4? [...]
>
> We changed the figure and used the same samples for the baseline and the experts. We also added another qualitative plot for Awa2 in figure 4.
>
> > [...] Pseudo code is given but no github implementations, so reproducibility may be cumbersome.[...]
>
> Please refer to the GitHub link in the general comment. Also, we listed all the hyperparameters in Appendix A.6.

---

> > ### Comment · Reviewer_1Nhn · 2022-11-18
> > **Thanks**
> >
> > Thanks for addressing my comments.
> >
> > I'm aware that your method is not post hoc explainability. Still, the aim is to have a prediction with an explanation, just as for post hoc explanation methods. I would think, in order to show that your method is superior to existing methods (and thus increased computational cost is justified), you would need to show that it doesn't compromise accuracy (which you did) and also that the explanations are of better quality than for post-hoc explainability.

---

> > > ### Author Response · Authors · 2022-11-19
> > > **Comparison with concept-based methods.**
> > >
> > > Thank you for engaging in discussion with us; we appreciate it.
> > >
> > > In response, we would like to point out the following:
> > >
> > > * Our method is closer to the interpretable approaches because it requires training a new model. We are following the experiment design in that category [1-6], where the evaluation is against a  BlackBox model, and the metric is how much the interpretable design hurts the classification performance. We already have that results (see Fig. 2). None of [1-6] compare against CAV because CAV is a post hoc explanation method (involving no training).
> > >
> > > > "... that the explanations are of better quality than for post-hoc explainability."
> > >
> > > * Arguably, the posthoc explanation methods can be divided into feature attribution (eg GradCAM) that generates feature importance heatmap and concept-based explanation. The feature attribution approaches are not applicable because they don't use concepts. The closer ones are concepts-based post hoc explanations such as CAV. We are not aware of any paper that provides a protocol to compare the quality of explanation between an interpretable method (like ours) and post hoc explanations. Among the post hoc explanations method, TCAV is used for comparison that requires a pre-trained model. This is not applicable to our method because our method requires training a new network. Furthermore, TCAV is criticized for not being a good metric to compare posthoc concept-based explanation (see [7], Fig 5).
> > >
> > > * One idea to compare between explanation/interpretable methods is to evaluate whether the explanation is *actionable* meaning if concepts are correctly identified, one should be able to intervene and *fix/alter* BlackBox's property. Our last experiment (Section 4.3) is designed for that purpose. We showed that one could use our method to fix the shortcut learning issue of a BlackBox. None of the posthoc approaches can do this because there is no mechanism to intervene in the BlackBox.
> > >
> > > * Finally, for the sanity check, we include the result of the random intervention on the concepts. Such intervention should degrade the performance of our model. For example, for CUB-200 VIT-derived MoIE, the performance of MoIE drops from 91.30 to 60.13% ($\Delta$=31.17%) (see Table 8 in the Appendix A.9). We did the same experiment for another concept-based method [1] (our baseline), and the drop is from 85.20% to 65.02 %  ($\Delta$=20.18%). Bigger $\Delta$ suggests more sensitivity wrt to the derived concepts.
> > >
> > >
> > > ## References
> > > [1] Concept Bottleneck Models, Koh et al.
> > >
> > > [2] Post-hoc Concept Bottleneck Models, Yuksekgonul et al.
> > >
> > > [3] Entropy-based Logic Explanations of Neural Networks, Barbiero et al.
> > >
> > > [4] Concept Embedding Models, Zarlenga et al.
> > >
> > > [5] Logic Explained Networks Ciravegna et al.
> > >
> > > [6] A Framework for Learning Ante-hoc Explainable Models via Concepts, Sarkar et al.
> > >
> > > [7] Concept Activation Regions: A Generalized Framework For Concept-Based Explanations, Crabbé et al.

---

### Official Review · Reviewer_7Y2K · 2022-10-24

**Confidence:** 5
**Correctness:** 1
**Technical Novelty And Significance:** 1
**Empirical Novelty And Significance:** 1
**Recommendation:** 1

**Clarity, Quality, Novelty And Reproducibility:**

+ Clear problem
- No novelty as state-of-the-art is missing cf. (1), and specially (2) which is basically capturing AND/OR expression for annotating neurons, and then could used for computing what has been described in the paper.

(1) David Bau, Bolei Zhou, Aditya Khosla, Aude Oliva, Antonio Torralba: Network Dissection: Quantifying Interpretability of Deep Visual Representations. CVPR 2017: 3319-3327
(2) Jesse Mu, Jacob Andreas: Compositional Explanations of Neurons. NeurIPS 2020




**Strength And Weaknesses:**

+ Interesting problem
- Clear missing state of the art cf. below
- Limited experiments - specially in the context of semantic segmentation

**Summary Of The Paper:**

This paper proposes a method to iteratively extract a mixture of interpretable models from a trained Blackbox

**Summary Of The Review:**

Overall the paper is well described, but novelty is not high given past works have achieved similar results

---

> ### Author Response · Authors · 2022-11-12
> **posthoc explanation vs interpretable method. We would be thankful if you could elaborate more.**
>
> Despite the brevity and lack of specificity of the review, we try to address comments to the best of our knowledge. We
> hope to engage with the reviewer to understand his/her/their concern and go beyond a superficial understanding of our
> paper.
>
> * It seems the main purpose of our manuscript and those of Bau et al and Mu et al are not well understood. Bau et al and
>   Mu et al aim at **posthoc explanation** of the black-box. Both papers stop at the explanation and do not result in a
>   new **interpretable network**. Please note that post hoc explanation and interpretable by design are two different
>   categories of eXplainable AI (XAI). Further more, Bau et al and Mu et al utilize different strategies than ours to capture the concepts.
> * This paper is about streamlining the development of a new interpretable model by taking advantage of the flexibility
>   of a black-box DL. In that sense, neither Bau et al nor Mu et al are SOTA for interpretable methods.
> * In the current literature of XAI, posthoc explanation and interpretable methods are two distinct design choices that
>   need to be decided at the beginning. We aim to blur that line.
> * About novelty: please see the general comment about novelty.

---

> > ### Comment · Reviewer_7Y2K · 2022-12-02
> > **Further point-wise comparison would be still valuable**
> >
> > Many thanks for the answers. I agree the approaches do no fall in the same category of XAI techniques. However concepts aims being captured in both work, which make them closer and comparable even if there are classified differently. I strongly think the paper would benefit from a clear point by point comparison of the approaches - this will strongly help anyone in better understanding the differences between the concepts / semantics captured.
> >
> > I really think this is a strong stopper for publication as the work is not positioned with respect to works which fundamentally address very similar objectives.

---

> > > ### Author Response · Authors · 2022-12-03
> > > **Comparison using the concepts by Bau et al.**
> > >
> > > Thank you for raising this point in discussion with us; we appreciate it.
> > >
> > > In response, we would like to point out the following:
> > >
> > > * First Bau et al. aims to find concept-neuron association but we aim to find how the different concepts are composed
> > >   together for the class prediction. Though both of them deals with concepts, the fundamental objective of the two
> > >   methods are different. However, as suggested in [1, 2], we can leverage the concepts captured in BRODEN dataset by Bau
> > >   et al. to design an interpretable model for datasets where concept annotation is scarce. We first project the
> > >   representation of the blackbox $\Phi$ to the concept space spanned by the concept vectors in Broden dataset. Next, the
> > >   interpretable model consumes the projected concept vectors to predict the class labels. To do so, we create a concept bank
> > >   using the concepts from BRODEN dataset following [1, 2]. We capture a total of 170 visual concepts as mentioned in
> > >   table 5 in [1] like 'hair' or '
> > >   eyebrow'. As CIFAR10 and CIFAR100 datasets does not have an explicit concept annotation, we utilize these concepts to
> > >   design the Mixture of Interpretable experts (MoIE) for image classification. As a blackbox, we use CLIP-ResNet50 as
> > >   referred in [2]. We follow similar hyperparameter settings as in [1, 2]. Refer below for the comparison of Top-1
> > >   Accuracy of the different models.
> > >
> > > | Method                             | CIFAR10 | CIFAR100 |
> > > |------------------------------------|---------|----------|
> > > | Blackbox (ClipResNet50)            | 87.6 %  | 68.1 %   |
> > > | MoIE (99 % coverage)               | 85.4 %  | 66.3 %   |
> > > | Interpretable by design (Baseline) | 83.2 %  | 62.8 %   |
> > >
> > > ## References
> > >
> > > [1] Meaningfully Debugging Model Mistakes using Conceptual Counterfactual Explanations, Abid et al.
> > >
> > > [2] Post-hoc Concept Bottleneck Models, Yuksekgonul et al.

---

> > > > ### Author Response · Authors · 2022-12-09
> > > > **Did we answer your question?**
> > > >
> > > > Dear Reviewer,
> > > >
> > > > In the remaining time, we would be thankful if you let us know if the experiment was convincing or if you need different types of experiments/clarification.
> > > >
> > > > If satisfied, we would be thankful if you update your score accordingly.

---

### Official Review · Reviewer_sRwr · 2022-10-26

**Confidence:** 3
**Correctness:** 3
**Technical Novelty And Significance:** 3
**Empirical Novelty And Significance:** 3
**Recommendation:** 8

**Clarity, Quality, Novelty And Reproducibility:**

Clarity
The manuscript has a good flow and its content is relatively easy to follow. Overall it was a good read.

Quality
The quality of the manuscript is good

Novelty
While the idea of using a surrogate model to enable the explanation of an existing model is not new, the way in which the proposed method implements that idea has several aspects that I find novel.

Reproducibility
The presentation of the method is relatively clear. Unless I overlooked something, I would not expect major challenges while trying to reproduce the reported results.

**Strength And Weaknesses:**

Strengths
+ well-supported idea
+ tested for bias removal
+ Good presentation

Weaknesses
- Relevant aspects not evaluated
- Limited novelty
- Some open questions

**Summary Of The Paper:**

The manuscript proposes to bridge the gap between post-hoc explanation methods and interpretable-by-design methods.

This is achieved by a pipeline where given a black-box model, an interpretable component iteratively distills parts of the representation from the blackbox with uncovered (undistilled) parts forwarded to a residual component. By iterating on this process, the interpretable components achieves a higher coverage of the blackbox model while still preserving a competitive performance.

Experiments on several datasets show the effectiveness of the proposed method on the image classification task for a variety of datasets. In addition, an alternative application of the method is presented on the task of bias detection.


**Summary Of The Review:**

The proposed idea is sound and most of the design decisions are well motivated. To the best of my knowledge, the idea of having this iterative combination of interpretable components with residual components is novel.

I also found positive the fact that the proposed method aims at interpreting different levels of the blackbox model and does not limit itself to the last convolutional layer, as is commonly the case.

The presentation of the manuscript is good, and the structure followed for the presentation of its content is adequate.

My main doubts with the manuscript are the following:

- Positioning with respect to Prototype-based methods. In Section 2, the paper discusses ProtoPNet as a representative of prototype-based methods. As part of this discussion, weaknesses from ProtoPNet are stressed. My concern here, is that ProtoPNet is nothing more than the first of these family of prototype-based methods. In this regard there are more recent variants, e.g. Proto-Tree (Nayuta et al., CVPR21) which have addressed the weaknesses of the original ProtoPNet model. Here I would expect the positioning to be made with respect to more recent prototype-based methods.

- On the selection of the number of experts. When describing the training configuration, the number of experts used for each blackbox-dataset combination was indicated. However, it is not clear to me the basis on which the number of experts is selected. Is there a principle manner to decide on the number of experts to use for the proposed method?

- On the semantic association of the detected concepts. In Section 4.2.2, local explanation produced by the MoIE are discussed. Several of these are described based on concepts with a clearly associated semantic meaning , e.g. IrregularStreaks, IrregularDG and Blue Whithish Veil. H

- On the explanation capabilities of the proposed method. Independently of its inner-workings, at the end of the day the proposed method is an explanation method. In this regard, it would have strengthen the manuscript if the proposed method was tested under the sanity checks proposed by [Adebayo et al., NeurIPS'18]. This way there could be guarantees on whether the generated heatmaps/attribution maps do constitute valid explanations.

- Quantitative Comparison. At the moment there is no quantitative comparison on the outputs of the proposed methods with respect to those from state of the art methods. In this regard, quantitatively speaking, it is hard to assess whether the proposed method is a better explanation method than existing ones.

- Minor. There are typos in several places of the manuscript.

--------------
References

 Meike Nauta, Ron van Bree, Christin Seifert; Proceedings of the IEEE/CVF Conference on Computer Vision and Pattern Recognition (CVPR), 2021, pp. 14933-14943

---

> ### Author Response · Authors · 2022-11-12
> **posthoc explanation vs interpretable method, comparison with ProtoTree, Novelty, and stoping principle**
>
> We thank the reviewer for the constructive and thoughtful comments. We hope that the following discussion addresses your
> questions.
>
> * Please find the comment about the novelty above.
>
> > [...] Proto-Tree (Nayuta et al., CVPR21) which have addressed the weaknesses of the original ProtoPNet model. Here I would expect the positioning to be made with respect to more recent prototype-based methods. [...]
>
> We agree with the reviewer that the prototype method is one of the papers in that family and indeed the ProtoTree
> addresses some of those issues. We will add that citation to the paper. Our method offers a more flexible interpretable
> method that improves accuracy (Table below for CUB-200 dataset).
>
> | Method                                   | Top-1 Accuracy |
> |------------------------------------------|---------------|
> | ProtoPNet, Chen et al. [Neurips, 2018]   | 79.2 %        |
> | ProtoTree h=9, Nayuta et al.[CVPR, 2021] | 82.2 %        |
> | MoIE (ours, ResNet Backbone)             | 88.6 %        |
> | MoIE (ours, VIT Backbone)                | 91.3 %        |
>
> There are also the following key differences:
>
> - Our method allows leveraging a blackbox and distilling it to any symbolic method (including ProtoTree), while a
>   Prototype-based approach should be trained from scratch. Training from scratch can be a difficult optimization task,
>   depending on the template or architecture of the interpretable method.
> - The samples routed to the last residuals can be viewed as a subset of data for which the template of the interpretable
>   method is not appropriate. Neither Prototype nor ProtoTree offers such flexibility.
> - Using prototype approaches to fix undesirable properties such as shortcuts is not straightforward. We have shown that
>   our method can easily be used for such applications.
> - Our method is also tested on a more diverse dataset.
>
> > Is there a principle manner to decide on the number of experts to use for the proposed method? [...]
>
> We follow two principles to stop the recursive process. 1) Each expert should have enough data to be trained reliably (
> coverage $\zeta^k$). If insufficient samples are assigned to an expert, we stop the process. 2) If the latest residual (
> $r^k$) is underperforming, it is not a reliable black box to distill from. We stop the procedure to avoid degrading the
> overall accuracy. We add a section about stopping criteria in the appendix or in the algorithm block in **red** color.
>
> > [...] In this regard, it would have strengthen the manuscript if the proposed method was tested under the sanity checks proposed by [Adebayo et al., NeurIPS'18]. [...]
>
> Adebayo et al. is about sanity check for the posthoc explanation method and, more specifically saliency-based approach.
> Our method is close to the category of interpretable methods. Unlike the traditional interpretable method trained from
> scratch, we use the flexibility of a blackbox DL to design an interpretable method on-the-fly (during training)
> progressively.
>
> > [...] Quantitative Comparison At the moment there is no quantitative comparison on the outputs of the proposed methods with respect to those from state of the art methods [...]
>
> This comment is related to the previous one. Our method is **not** a post hoc explanation method but rather an
> interpretable approach. Many quantitative experiments in the paper are consistent with the evaluation of the
> interpretable methods:
>
> - **Does our method compromise accuracy?** Sec 4.2.1 and Figure 2 show that our method outperforms the interpretable
>   baseline (Concept bottleneck model Koh et el. 2020) and does not compromise the performance of the respective
>   blackbox.
>
> - **Is there any sample that does not fit the template of the interpretable method?** There are always samples that need
>   to fit the interpretable template. Our last selector ($g^k$) can identify those. Note that for these samples the
>   performance of the blackbox should be low. We compare the performance of the black box for these samples with the last
>   residual ($r^k$). Please refer to section A 7.7 and Table 6 in the supplementary material for details.
>
> - **Is this a helpful method for application?** This is the primary concern about any XAI approach. We showed that our
>   method could fix the shortcut bias. Please refer to section 4.3 for details.
>
> > [...] On the semantic association of the detected concepts In Section 4.2.2, local explanation produced by the MoIE are discussed. Several of these are described based on concepts with a clearly associated semantic meaning , e.g.
>
> It seems that the last sentence is missing. We would be thankful if you update your comment.

---

> > ### Comment · Reviewer_sRwr · 2022-11-16
> > **thanks for the feedback**
> >
> > I thank the authors for addressing my initial comments in their rebuttal.
> > To a good extent the provided feedback has clarified some of my initial doubts.
> > I would encourage the authors to make an effort so that these additional discussions and clarification pointers make it to the main manuscript.
> >
> > My apologies for the incomplete paragraph, thanks for pointing it out.
> > I am adding the complete version below:
> >
> > "= On the semantic association of the detected concepts:
> > In Section 4.2.2, local explanation produced by the MoIE are discussed. Several of these are described based on concepts with a clearly associated semantic meaning , e.g. "IrregularStreaks", "IrregularDG" and Blue Whithish Veil". However, unless I am overlooking something, it is not clear to me how the connection is made with respect to these concepts. Are these concepts additional annotations in the dataset?"
> >
> > Overall my impression of the manuscript remains positive.
> > I will have a look at the concerns raised by other reviewers and the provided feedback to those concerns.
> > Based on that, and the response to the previous comment, I might consider increasing my initial review.

---

> > > ### Author Response · Authors · 2022-11-16
> > > **Details on the concept annotations**
> > >
> > > Thank you for the comment. Please refer to our response below:
> > >
> > > > [...]I would encourage the authors to make an effort so that these additional discussions and clarification pointers make it to the main manuscript? [...]
> > >
> > > We already incorporated all the review comments from the 1st round. We will collect more feedback from reviewers and modify
> > > the paper accordingly. Please refer to Appendix A.7.3 for the comparison with the prototype-based methods.
> > >
> > > > [...] I am overlooking something, it is not clear to me how the connection is made with respect to these concepts. Are these concepts additional annotations in the dataset? [...]
> > >
> > > We handle both the cases when we have and don't have the explicit annotations for the concepts in the data. For datasets
> > > like CUB-200 and Awa2, we have the concept annotation in the data in the form of attributes. For skin dataset, HAM 10k
> > > we acquire the concept annotations from derm7pt dataset [1, 2]. For chestXrays, we use the Stanford RadGraph pipeline to
> > > obtain the anatomical and observation concepts from the radiology reports [3]. We mention all our assumptions in the
> > > method section 3 (Notation). Furthermore, we describe all the datasets in details in the Appendix section A.1 (Datasets)
> > > .
> > >
> > > References:
> > >
> > > 1. Lucieri et al. On interpretability of deep learning based skin lesion classi- fiers using concept activation vectors.
> > >    In 2020 international joint conference on neural networks
> > >    (IJCNN), pp. 1–10. IEEE, 2020.
> > > 2. Daneshjou et al. Disparities in dermatology ai: Assessments using diverse clinical images. arXiv preprint arXiv:
> > >    2111.08006, 2021
> > > 3. Yu et al. Anatomy-guided weakly-supervised abnormality localization in chest x-rays. arXiv preprint arXiv:2206.12704,
> > >    2022

---

### Author Response · Authors · 2022-11-12
**General comment about Novelty and code**

We thank all reviewers for their feedback. We address each reviewer separately. In the following, please find our general comment.
## Code:
For the code, please refer to the [github-link](https://github.com/annonymousai/Route-interpret-repeat-iclr). We uploaded the code for ResNet-101 based model for CUB dataset and the baseline. We will upload the code for the other datasets upon decision.

## About Novelty:
  - Our method is the first to blur the distinction between **post hoc** explanation and **interpretable** by design-based
  approaches by **carving an interpretable model out of the BlackBox**. The method shows how to maintain interpretability without compromising accuracy by gradually distilling from a BlackBox and building a new hybrid symbolic-Blackbox until convergence.  **Our method is not just a mere post-hoc explanation technique like saliency maps or concept activation.** To the best of our knowledge, there is no such approach in the literature.
- We showed that the number of concepts to explain a sample can vary based on the architecture of the network (CNN and VIT).
 - We showed that by taking advantage of a BlackBox (distilling), we achieved better performance than training from scratch (classical interpretable by design method).
  - There is always a subset of samples for which the template of the interpretable architecture is not optimal. Our approach can identify those via the last selector ($\pi^k$).
- Our method can be viewed as a framework, allowing any symbolic backbone for the interpretable method, making our approach applicable to a wide range of data types and applications (image, language, tabular data, video). We showed results using First Order Logic (FOL), but many other choices are possible.

---

### Decision · Program_Chairs · 2023-01-20

**Decision:**

Reject

**Justification For Why Not Higher Score:**

The reviewers remain unconvinced about the value and novelty of this approach in the context of prior work despite a lengthy discussion with the authors

**Justification For Why Not Lower Score:**

NA

**Metareview: Summary, Strengths And Weaknesses:**

This paper proposes a method for making a pre-trained NN model more interpretable by applying an iterative process of distilling an interpretable component from the model through training, hence splitting the model into an interpretable part and a residual black-box, and applying this process to the black box until no further distillation is possible. The reviewers agree that the paper is well written and the performance of the approach is evaluated well through extensive experiments. Saying this, the reviewers agree that the paper is missing baseline comparisons with other relevant work in this area. Even though the authors have argued that their work is uniquely placed between post-hoc interpretability methods and interpretable-by-design approaches, and hence cannot be meaningfully compared to any existing baselines from the respective literature, the reviewers remained unconvinced and still wanted to see better positioning of this work in the context of prior work.

**Summary Of Ac-Reviewer Meeting:**

It was a borderline paper after the original round of reviews, but after I prompted the reviewers to engage further with the discussion, the scores converged towards rejection.